

# Carbon monoxide cycling in the Ria Formosa Lagoon (southern Portugal) during summer 2021

Guanlin Li[1], Damian L. Arévalo-Martínez[1, 2], Riel Carlo O. Ingeniero[1], Hermann W. Bange[1]

[1]Marine Biogeochemistry, Helmholtz Centre for Ocean Research Kiel, Düsternbrooker Weg 20, 24105 Kiel, Germany
[2]now at the Department of Ecological Microbiology, Radboud University Nijmegen, The Netherlands

*Correspondence to*: Guanlin Li (guli@geomar.de)

**Abstract.** Carbon monoxide (CO) is an atmospheric trace gas that plays a crucial role in the oxidizing capacity of the Earth's atmosphere. Moreover, it functions as an indirect greenhouse gas, influencing the lifetimes of potent greenhouse gases such as methane. Albeit being an overall source of atmospheric CO, the role of coastal regions in the marine cycling of CO and how
its budget can be affected by anthropogenic activities, remain uncertain. Here, we present the first measurements of dissolved CO in the Ria Formosa Lagoon, an anthropogenically influenced system in southern Portugal. The dissolved CO concentrations in the surface layer ranged from 0.16 to 3.1 nmol L$^{-1}$ with an average concentration of 0.75 ± 0.57 nmol L$^{-1}$. The CO saturation ratio ranged from 1.7 to 32.2, indicating that the lagoon acted as a source of CO to the atmosphere in May 2021. The estimated average sea-to-air flux density was 1.53 µmol m$^{-2}$ d$^{-1}$, mainly fueled by CO photochemical production. Microbial consumption
accounted for 83 % of the CO production, suggesting that the resulting CO emissions to the atmosphere were modulated by microbial consumption in the surface waters of the Ria Formosa Lagoon. The results from an irradiation experiment with aquaculture effluent water indicated that aquaculture facilities in the Ria Formosa Lagoon seem to be a negligible source of atmospheric CO.

## 1 Introduction

Carbon monoxide (CO) plays an important role in the oxidizing capacity of the Earth's atmosphere. In the troposphere, CO rapidly reacts with hydroxyl radicals (OH), and in the presence of NO$_x$, it leads to the formation of ozone, a pollutant, and a greenhouse gas (Thompson, 1992). Moreover, due to its reactivity with OH, CO indirectly influences the lifetime of atmospheric methane, a potent greenhouse gas (GHG) (Canadell et al., 2021). Consequently, CO is an important indirect greenhouse gas with an effective radiative forcing comparable to nitrous oxide (Forster et al., 2021).


The open and coastal oceans contribute up to 1 % of the global atmospheric CO sources (Zheng et al., 2019). Current estimates indicate that oceanic emissions of CO may range between 9 Tg CO yr$^{-1}$ and 20 Tg CO yr$^{-1}$ (Stubbins et al., 2006; Park and Rhee, 2016; Conte et al., 2019; Zhang et al., 2019), while some early estimates (published before 1999) indicate oceanic CO emissions of up to 1200 Tg CO yr$^{-1}$ (see overview in Zafiriou et al., 2003). Coastal areas appear to be 'hot spots' of CO
emissions. Park and Rhee (2016) estimated that about 15–25 % of the global oceanic CO emissions can be attributed to coastal regions. The primary source of CO is its photochemical production via photo-oxidation of chromophoric (colored) dissolved





organic matter (CDOM) in the open and coastal oceans (e.g., Zafiriou et al., 2003; Ossola et al., 2022). Production by phytoplankton and the so-called 'dark production' are comparatively small but significant sources of oceanic CO (Zhang et al., 2008; Gros et al., 2009; McLeod et al., 2021). Microbial consumption is the major CO loss mechanism, accounting for

about 80–90 % of the CO loss from the marine environment (Zafiriou et al., 2003; Moran and Miller, 2007; Greening and Grinter, 2022). Therefore, CO emissions to the atmosphere are a minor sink of oceanic CO (Zafiriou et al., 2003).

The Ria Formosa Lagoon system is an anthropogenically influenced coastal area in southern Portugal along the Northeast (NE) Atlantic Ocean coast. Due to its restricted (estuarine-type) circulation and high nutrient and organic matter inputs, the Ria

Formosa Lagoon system could be a potential source of atmospheric CO. In this study, we present the first in situ observations of CO in the region, which we combined with incubation experiments in waters directly influenced by anthropogenic activities. The main objectives of this study were to i) elucidate the CO distribution and sea–air flux densities in the coastal lagoon system, ii) estimate the different CO sources and sinks in the lagoon system during summer, and iii) determine the potential impact of aquaculture activities on CO cycling in this region.

## 2 Study site description

Ria Formosa is a major lagoon system in southern Portugal (36°58' N, 8°02' W to 37°03' N, 7°32' W) (Newton and Mudge, 2003; Cravo et al., 2014). It is a mesotidal coastal lagoon with an average water depth of ~2 m. Water temperatures typically range from about 12 °C in the winter to about 27 °C in the summer (Newton and Mudge, 2003). Salinities range from 13 to 36.5 (Newton and Mudge, 2003). The lagoon is vertically well-mixed due to limited freshwater inputs and the predominance

of a tidal-forced circulation pattern (Cravo et al., 2014). Being a typical coastal lagoon system, the water exchange with the adjacent shelf is restricted (Tett et al., 2003). In addition to nutrient inputs from aquaculture facilities, the Ria Formosa Lagoon receives significant nutrient inputs from agricultural runoff and sewage, particularly from the cities of Faro, Olhão, and Tavira (Newton and Mudge, 2005). A part of the Ria Formosa Lagoon system is a natural park established in 1987 (Aníbal et al., 2019). Ria Formosa also plays a vital role in the region's tourism industry.

## 3 Materials and methods

### 3.1 Sample collection

Discrete water samples for determining CO concentrations were collected using a Niskin bottle at approximately 1 m water depth from a small boat at 14 stations in the western part of the Ria Formosa Lagoon (Fig. 1). Samples were taken in the late afternoon (~17:00h local time) on 25 May 2021. The sampling was repeated at approximately the same time on 26 May 2021.

The sampling time coincided with the high tide period of the semi-diurnal tidal cycle. Additionally, atmospheric samples for CO were collected at Stations 8, 9, 10, 11, 12, and 14 (Fig. 1) on 26 May 2021.



Bubble-free water samples were collected as triplicates for the determination of dissolved CO concentration and were transferred into 100 mL quartz glass bottles from the Niskin bottle. Although CO sample uptake within the vial at ambient conditions was noted by Day and Faloona (2009), our samples were stored in cooling boxes to avoid warming immediately after sampling and transported within 1.5 hours to the Centro de Ciências do Mar (CCMAR) for analysis (our measured CO loss is less than 9% under the same storage conditions (Fig. S1)).

Triplicate atmospheric CO samples were collected windward of the boat by using 20 mL gas-tight glass syringes when the boat stopped for station work.

Water samples from Station 7 (the representative area for the combined effects of tidal effects and terrestrial contributions) for determining CO consumption were introduced into 100 mL quartz glass bottles from the Niskin bottle and then covered with aluminum foil and stored in a cooling box after sampling. The sample at time zero (the starting point) of the consumption experiment was poisoned with saturated aqueous sodium azide ($NaN_3$) solution immediately after sampling. All samples for the consumption experiment were transported back to the laboratory within half an hour. The second data point was analyzed 30 minutes after sampling. The rest of the samples were measured every 30 to 60 min. Generally, measurements at five time points were performed during the duration of the microbial CO consumption experiment (see Section 4.3)

### 3.2 CO measurements

CO in the seawater and atmosphere was detected using a CO Gas Analyzer (ta3000R; Ametek, USA). Calibration of the instrument was performed every half hour by using a standard gas mixture with a CO mole fraction of 113.9 ppb CO in synthetic air (DEUSTE Gas Solutions, Germany), which was calibrated against a certified standard gas (250.5 ppb CO, calibrated against the NOAA 2004 scale at the Max Plank Institute for Biogeochemistry, Jena, Germany).

Gas samples were injected into the gas analyzer using a gas-tight Luer-lock syringe. Equilibrated atmospheric CO mole fractions were measured using the headspace method described in Campen et al. (2023) and Xie et al. (2002). The headspace of the seawater samples was created by replacing 12 mL of water with CO-free gas (zero gas). The zero gas was produced by removing CO from a commercially available synthetic air gas mixture with CO ≤ 1 ppm (Alphagaz 2, AirLiquide, Germany) with a CO-scrubber (MC1-906FV; SAES Pure Gas, USA). The samples were equilibrated for 5 minutes at 120 rpm on a KS-10 shaking table (Edmund Bühler, Germany), followed by an additional equilibration period of 3 min before analyzing a gaseous subsample of the headspace with the CO analyzer.

Concentrations of dissolved CO in seawater ($CO_{surf}$ in nmol $L^{-1}$) were calculated as:



$$CO_{surf} = x' \times \beta + \left(\frac{x' \times P}{R \times T} \times \left(\frac{V_a}{V_w}\right)\right) \qquad (1)$$

where β is the Bunsen coefficient of CO, which was computed as a function of temperature (T) and salinity (S) (Wiesenburg and Guinasso, 1979), x' is the CO dry mole fraction, P is the atmospheric pressure at the time of equilibration (or time of the atmospheric sampling), $V_w$ is the volume of the water sample (in mL), $V_a$ is the volume of the headspace air (in mL), R is the molar gas constant 0.08206 atm L mol$^{-1}$ K$^{-1}$, and T is the equilibration temperature in K. The CO dry mole fractions were computed according to Weiss and Price (1980):

$$x' = x/\left(1 - \frac{P_{H_2O}}{P}\right) \qquad (2)$$

$x$ is the measured CO moist mole fraction, and $P_{H2O}$ is the partial pressure of H$_2$O at the time of equilibration (or time of atmospheric sampling). ln(P$_{H2O}$) as a function of T and S can be calculated according to Eq. 3 in Weiss and Price (1980):

$$In(P_{H_2O}) = 24.4543 - 67.4509 \times \left(\frac{100}{T}\right) - 4.8489 \times In\left(\frac{T}{100}\right) - 0.000544 \times S \qquad (3)$$

The dissolved CO concentration (CO$_{eq}$ in mol L$^{-1}$) in equilibrium with seawater was calculated using (Wiesenburg and
Guinasso, 1979):

$$CO_{eq} = x' \times \beta \qquad (4)$$

Because of a technical problem with the calibration of the CO analyzer, we corrected our measurements according to:

$$x' = f \times x'_0 \qquad (5)$$

where $x'_0$ is the measured dry mole fraction and f is the correction factor of 3.12. f, which was computed as the ratio of mean
atmospheric CO mole fractions from the atmospheric monitoring stations Mace Head (Ireland) and Terceira (Azores) in May 2021 (109.5 ppb; Petron et al., 2022) to the mean of our atmospheric measurements (35.1 ± 5 ppb). The atmospheric lifetime of CO, ranging from approximately 1–4 months (IPCC, 2021), is sufficiently long to yield nearly uniform atmospheric background CO mole fractions across the Northwestern Atlantic Ocean during our measurement period. Thus, atmospheric CO mole fractions measured at Mace Head and Terceira can be considered representative for the Ria Formosa Lagoon region.
For each set of triplicate CO concentration measurements, we calculated the standard error. The overall mean error for the dissolved CO measurements was ± 2.7 %.

### 3.3 Calculation of CO saturations and flux densities

The saturation ratio of CO, α, was computed as:

$$\alpha = CO_{surf}/CO_{eq} \qquad (6)$$

The sea–air flux densities of CO, $F_{sea–air}$, were calculated by using:



$$F_{sea-air} = k_{CO}(CO_{surf} - CO_{eq}) \tag{7}$$

Here the gas transfer coefficient $k_{CO}$ was calculated with the approach of Raymond and Cole (2001), which was established for estuaries and rivers:

$$k_{CO} = (1.91\, e^{0.35u_{10}}) \cdot (Sc_{CO}/660)^{-n} \tag{8}$$

with n = 2/3 for wind speeds in 10 m height ($u_{10}$) < 3.7 m s$^{-1}$ and n = 1/2 for $u_{10}$ > 3.7 m s$^{-1}$. In this study, wind speeds in 10 m height were parameterized assuming the power law representation (Panofsky and Dutton, 1984):

$$\frac{u_{10}}{u_2} = \left(\frac{10}{2}\right)^{\gamma} \tag{9}$$

where $\gamma$ is an empirical coefficient equal to 0.094855.

The Schmidt number ($Sc_{CO}$) of CO was computed as in Zafiriou et al. (2008) with temperature ($T$) in Celsius degrees:

$$Sc_{co} = -0.0553T^3 + 4.3825T^2 - 140.07T + 2124 \tag{10}$$

**3.4 Incubation experiment**

We performed an incubation experiment using waters from the effluent of the aquaculture facility *Estação Piloto de Piscicultura em Olhão* (EPPO) at the *Instituto Português do Mar e da Atmosfera, I. P.* (IPMA, I. P.) in Olhão (Fig. 1). Seawater
was collected in 3.5 L UV-transparent incubation bottles (DURAN®, quartz glass, GL 45, DWK Life Sciences, Germany) and placed into 200 L incubation enclosures (incubators) at CCMAR of the University of Algarve in Faro. The enclosures were constructed from plexiglass (GS 2458 UV transmitting) to allow the glass bottles to be exposed to the full natural sunlight spectrum. A constant flow of water (20 L min$^{-1}$) was maintained to minimize temperature deviations during the experiments (mean temperature: 25.4 ± 1.5°C). Bottles were incubated under light and dark conditions to investigate photochemical CO
production. A total of 16 bottles (separated equally) were distributed between two incubators. One incubator was set to light conditions (Group A), and the other was set to dark conditions (Group B). The experiments were carried out for 48 h, with sampling from the incubation bottles conducted at time points 0, 24, and 48 h. At each time point (14:00 h local time), triplicate samples were collected to determine dissolved CO concentrations and CDOM absorbance.

**3.5 Ancillary measurements**

Chlorophyll *a* (Chl-*a*), fluorescent dissolved organic matter (FDOM), salinity, pH, and water temperature (T) were measured on board in 0.5 m water depth using a calibrated EXO2 multiparameter sonde (YSI Inc.; Yellow Springs, Ohio, USA).



Water samples for nutrient analysis were collected in triplicate, filtered through cellulose acetate filters, and stored at -20 °C until analysis. The concentrations of ammonium ($NH_4^+$), nitrate ($NO_3^-$), and phosphate ($PO_4^{3-}$) in the seawater were determined
using standard colorimetric techniques (μMac-1000; Systea, Anagni, Italy) (Alexandre et al., 2012).

CDOM absorption in samples from the incubation experiment was measured using a Thermo Fisher Spectroquant® Pharo 300 UV-Vis Spectrophotometer ranging from 250 to 800 nm wavelength with a 1 nm interval (Loiselle et al., 2009).

Instantaneous wind speeds at 2 m above sea level were measured onboard during atmospheric sampling using a vane anemometer (Testo, Germany).

## 4 Result and discussion

### 4.1 Distribution of dissolved CO in the Ria Formosa Lagoon

CO concentrations ranged from 0.16 to 3.1 nmol $L^{-1}$ and showed a considerable daily variability (Fig. 2). Significantly
enhanced CO concentrations were measured at Stations 1 and 2, which were located close to a wastewater treatment plant (WWTP) (Fig. 1). However, the lowest CO concentration was also measured at Station 1. The CO concentrations from the Ria Formosa Lagoon were considerably lower than the CO concentrations reported from the Big Lagoon in California (USA), which ranged from 100 to 1000 nmol $L^{-1}$ (Butler et al., 1988). The CO concentrations reported here are at the lower end of CO concentrations measured in shallow coastal and estuarine waters such as the Ishikari Bay (Hokkaido, Japan), the Jiaozhou Bay
(China), the Schelde Estuary (Netherlands) and the Yaquina Bay (Oregon, USA) which range from 0.3 to 50 nmol $L^{-1}$ (Butler et al., 1987; Law et al., 2002; Nakagawa et al., 2004; Wang et al., 2017). For comparison, CO concentrations in shelf and coastal upwelling waters range from 0.1 to 18 nmol $L^{-1}$ (see overview in Liss et al., 2014; Zhao et al., 2015; Zhang et al., 2019). The low concentrations reported here are partly caused by the fact that no diurnal cycles of dissolved CO were measured. Therefore, the time of maximum CO (photochemical) production might have been missed.


The mean concentrations (i.e., the average from the two sampling days) of dissolved CO and water temperature, salinity, Chl-a, FDOM, pH, and nutrients in the Ria Formosa Lagoon are shown in Fig. 3 and listed in Table 1. Due to sea salt production facilities and aquaculture ponds, the study area's salinity (mean salinity = 36.54) was higher than the seawater outside the lagoon. Chl-a concentrations (used here to estimate phytoplankton biomass) ranged from 0.02 to 0.50 μg $L^{-1}$, with an average
of 0.23 ± 0.16 μg $L^{-1}$. In general, Chl-a concentrations decreased from the Ramalhete to the Faro–Olhão inlet (Aníbal et al., 2019). in correspondence to increasing distance from the nutrient-rich plumes from the WWTP. High Chl-a concentrations were recorded at Station 8, indicating higher phytoplankton biomass (0.50 μg $L^{-1}$).





To assess the spatial variability in the lagoon, we divided the sampling area into three regions according to FDOM vs. CO, Sal vs. T plots (Fig. 4): WWTP plume zone (Stations 1 to 7), Praia de Faro zone (Station 8), and Faro–Olhão inlet zone (Stations 9 to 14). The WWTP plume zone was characterized by high temperature, low salinity, high nutrient ($NH_4^+$ and $PO_4^{3-}$) concentrations, and large amounts of FDOM. CO concentrations in this zone were strongly affected by environmental conditions with anthropogenic activities. Such effluents might be a source of FDOM, which could potentially enhance CO production (Zepp, 2003; Yang et al., 2011; Zhao et al., 2015). Contrarily, in the semi-diurnal tidal influenced Faro–Olhão inlet zone, a low-temperature, low-salinity, and low-biomass regime caused by "fresh" seawater entering the lagoon through the Faro–Olhão inlet could be observed. Consequently, close to this inlet, CO concentrations were comparatively low. Lastly, in the seagrass (*Cymodecea* and *Zostera*) region of the lagoon and in the Praia de Faro zone, where the highest phytoplankton biomass was observed, high CO concentrations were measured at Stations 8 and 13. This suggests a potential for direct biological CO production, which might be a significant source (Gros et al., 2009; McLeoad et al., 2021). Besides, during the high tide, the water of the Faro channel caused the WWTP thermal effluent plume to have a visible influence from the region of Montenegro through Esteiro Largo and to the West region through the Ramalhete (Aníbal et al., 2019).

Indeed, a regression analysis revealed that CO concentrations were significantly correlated with FDOM ($r = 0.416$, $p < 0.05$, $n = 28$). Since no significant correlations were found between CO concentrations and pH, water temperature, salinity, and Chl-*a*, it appears that the distribution of dissolved CO in the Ria Formosa Lagoon was mainly driven by the photochemical production of CO via FDOM at the time of sampling. Nonetheless, a significant correlation was found between the CO and $PO_4^{3-}$ concentrations ($r = 0.860$ $p < 0.01$, $n = 14$).

**4.2 Sea–air flux densities of CO**

The waters of the Ria Formosa Lagoon were supersaturated with CO at all stations, with saturation ratios ranging from 1.7 to 32.2 (mean ± SD: 7.7 ± 5.9; Fig. 5(a)), which are directly proportional to CO concentrations in the surface water ($r = 0.995$, $p < 0.01$, $n = 28$). This indicates that the lagoon was a source of atmospheric CO in May 2021, which concurs with the observations from other coastal waters (e.g., Liss et al., 2014).

The mean flux density of CO was estimated to be $1.53 ± 0.92$ µmol m$^{-2}$ d$^{-1}$ for the Ria Formosa Lagoon. The sea-to-air flux was lower than those obtained from the North Atlantic Ocean ($2.2 ± 1.5$ µmol m$^{-2}$ d$^{-1}$) and the Northern Hemisphere Oceans ($1.9 ± 1.3$ µmol m$^{-2}$ d$^{-1}$) (Stubbins et al., 2006) but roughly in the same range of flux densities reported by Zhang et al. (2019) with an average of 1.76 µmol m$^{-2}$ d$^{-1}$ and by Zhao et al. (2015) with an average of 1.78 µmol m$^{-2}$ d$^{-1}$.

As shown in Fig. 5 (b), the flux densities of CO exhibited considerable spatial variability. Much of the variability was caused by the wind speed, as the highest flux densities were computed for Station 14 (2.87 µmol m$^{-2}$ d$^{-1}$) and coincided with the highest wind speed (6.1 m s$^{-1}$), whereas the lowest flux densities were computed for Station 9 (0.53 µmol m$^{-2}$ d$^{-1}$) and coincided





with low wind speeds (3.3 m s$^{-1}$). However, at Station 8, which had the lowest wind speeds (3.1 m s$^{-1}$), the flux density of CO (1.09 μmol m$^{-2}$ d$^{-1}$) was high due to the higher concentration of dissolved CO (0.88 nmol L$^{-1}$) and higher saturation ratio of CO (8.2). Therefore, the large variability in the CO flux densities might also have been caused by spatial and temporal

variations in anthropogenic and biogenic sources of CO.

### 4.3 Microbial CO consumption

To investigate the microbial consumption of dissolved CO, we determined the CO consumption rate using the dark incubation method (Zafiriou et al., 2003; Zhang et al., 2019; Sugai et al., 2020). CO concentrations decreased sharply with time during incubation, following the first-order kinetics (y = 1.5084e$^{-0.403x}$, r$^2$ = 0.9992) (Fig. 6). This is consistent with previous results,

which reported an exponential decrease in CO concentrations with incubation time at locations such as the Delaware Bay, Northwest Atlantic, and East China Sea (Xie et al., 2005; Yang et al., 2010). The resulting CO consumption rate constant ($K_{bio}$) was 0.40 h$^{-1}$, comparable to published CO consumption rate constants ranging from 0.02 h$^{-1}$ to 1.11 h$^{-1}$ (Xie et al., 2005).

The relatively rapid turnover time (defined as 1/$K_{bio}$) of 2.5 h suggests that microbial CO consumption was the dominant

removal pathway of dissolved CO in the Ria Formosa Lagoon system in May 2021, significantly modulating atmospheric CO emissions. Multiplying $K_{bio}$ by the mean surface seawater CO concentration and the sampling depth (1 m) yields a rough estimate of the depth-integrated microbial consumption rate ($C_{bact}$) of 7.25 μmol m$^{-2}$ d$^{-1}$ in the surface layer for the study area.

### 4.4 Comparison of CO sources and sinks for Ria Formosa Lagoon in May 2021

The budget of CO for the surface layer (i.e., the sampling depth of 1 m) is driven by the balance of various sources and sinks,

such as microbial consumption ($C_{bact}$), air–sea gas exchange ($F_{sea–air}$), photochemical production ($P_{photo}$), dark production ($P_{dark}$) and production by phytoplankton ($P_{phyto}$). With the assumption of a steady state, the sum of the CO sources and sinks is equal to zero:

$$C_{bact} + F_{sea–air} \cong P_{photo} + P_{dark} + P_{phyto} \qquad (11)$$

$C_{bact}$ and $F_{sea–air}$ have been estimated (Sections 4.2 and 4.3). Because the Ria Formosa Lagoon is usually well-mixed, we

assumed that vertical mixing and advection were negligible at the time of our study.

$P_{phyto}$ can be roughly estimated with the CO production rates given in Gros et al. (2009) and the Chl-$a$ concentrations measured here. It is known that diatoms can contribute an overall average of 53 % of the total phytoplankton biomass in the Ria Formosa Lagoon system (Pereira et al., 2007). Hence, we estimated $P_{phyto}$ as:

$$P_{phyto} = P(cd) \times Chl-a \cdot h \cdot t_{light} \qquad (12)$$

where $h$ is the thickness of the sampling depth (set to 1 m), $P$(cd) is the maximum CO production rate given in Gros et al. (2009) (8.7×10$^{-4}$ μg CO (μg chlorophyll)$^{-1}$ h$^{-1}$ for the diatom *Chaetoceros debilis*), Chl-$a$ is the mean Chl-$a$ concentration measured during our study in May 2021 (0.23 μg L$^{-1}$) and the average daylight hours ($t_{light}$) in May is 14.20 h d$^{-1}$ in this area.



The resulting $P_{phyto}$ was $5.38\times10^{-5}$ µmol m$^{-2}$ d$^{-1}$. $P_{dark}$ was roughly estimated according to the parameterization of Zhang et al.
(2008) and Conte et al. (2019), in which $P_{dark}$ is estimated from CDOM (represented by its absorption at 350 nm), temperature
(T in °C), pH, and salinity as follows:

$$P_{dark} = a_{cdom}(350) \cdot \partial_{CO} \cdot h \cdot t_{dark} \tag{13}$$

$$In(\partial_{CO} \times 10^3) = -\frac{12305}{T} + 0.494 \cdot pH - 0.0257 \cdot S + 41.9 \tag{14}$$

where the $\partial_{CO}$ is $1.78\times10^{-2}$ nmol m L$^{-1}$ h$^{-1}$ (computed with the mean water temperature of 292.25 K, the mean pH of 8.14, and
the mean salinity of 36.54 at the time of sampling), the Station 7 mean a$_{cdom}$ (350) is 0.21 m$^{-1}$ and the average dark hours ($t_{dark}$)
in this area were 9.80 h d$^{-1}$ in May. The resulting mean $P_{dark}$ is $3.66\times10^{-5}$ µmol m$^{-2}$ d$^{-1}$.

Finally, the CO photoproduction rate, $P_{photo}$, was calculated as $C_{bact} + F_{sea–air}$ - $P_{dark}$ - $P_{phyto}$. The estimates of the production and
consumption rates are given in Table 2. It is apparent that both $P_{phyto}$ and $P_{dark}$ were negligible compared to $P_{photo}$ (Table 2).

The main sink of CO in Ria Formosa in May 2021 was the microbial consumption of CO ($C_{bact}$), which accounted for about
83% of the total CO loss terms ($C_{bact} + F_{sea–air}$). Our results align with the estimate of Zhang and Xie (2012), who reported that
the microbial consumption of CO in the St. Lawrence Estuary accounted for 86–92% of the CO sink. Overall, the CO emissions
to the atmosphere resulted from CO photoproduction, which, in turn, was modulated by microbial CO consumption. This
supports the perception that microbial CO consumption is the dominant sink of oceanic CO, recycling approximately 80–90%
of the CO in the open and coastal oceans before it is released into the atmosphere (e.g., Moran and Miller, 2007; Zafiriou et
al., 2003). Furthermore, our estimates suggest that the CO budget in the Rio Formosa Lagoon system is balanced, with sources
and sinks having roughly similar magnitudes.

## 4.5 CO production from aquaculture effluent

The results of the 48h-incubation experiments with waters from the effluents of the aquaculture facility *Estação Piloto de
Piscicultura em Olhão* (EPPO) are shown in Fig. 7. The samples exposed to light showed a linear increase in CO concentration
over 48 h. The CO concentrations in the samples kept in the dark showed a steep decrease until 24 h and then continued to
decrease only slightly until 48 h. This indicates that microbial CO consumption, probably balanced by CO dark production,
led to a steady state at the end of the dark incubation.

The absorption of CDOM during the light incubation experiments is shown in Fig. 8. Absorption by CDOM as a function of
wavelength a(λ) (m$^{-1}$) which is commonly modeled with an exponentially decreasing function (Shifrin, 1988) of the form:

$$a(\lambda) = A \times e^{-S\lambda} \tag{15}$$

where λ is a reference wavelength, *A* is amplitude, and *S* (nm$^{-1}$) is the spectral slope parameter describing the relative steepness
of the spectrum, understood here as a proxy for changes in the composition of CDOM (Carder et al., 1989).



The ratios $S_R$ of two different spectral slopes, $S_{280–295}$ and $S_{350–400}$, were used to indicate the magnitude of the relative molecular weight of CDOM (Helms et al. 2008). The ranges of variation were 0.013 to 0.02 nm$^{-1}$ and 0.015–0.036 nm$^{-1}$ for $S_{280–295}$ and $S_{350–400}$, respectively, at different incubation times (Table 3). Further, we obtained $S_R$ mean values for different incubation times ($S_R$-0h: 0.56; $S_R$-24h: 0.52; $S_R$-48h: 1.07). The $S_R$ value is a measure of CDOM that provides insight into the sources and transformations of CDOM in the water column, independent of its concentration (Twardowski et al., 2004). Variations indicate changes in CDOM degradation during light incubations, potentially due to alterations in the CDOM composition and the ratio of fulvic acids to humic acids (Carder et al., 1989). A lower $S_R$ indicates a predominant terrestrial input with a typically higher proportion of humic acid and a higher photochemical reactivity potential. CO is mainly produced by the photodegradation of specific compounds (e.g., humic acid), which are efficient CO producers (Ren et al., 2014). In the initial phase of the experiment (0–24 h), specific aromatic compounds were destroyed, and their concentrations decreased rapidly under the consumption of photodegradation; as the experimental time grew (24–48 h), the concentration of the second type of precursors without humic acid gradually dominated, making the CO photoproduction rate decrease continuously. Notably, the microbial degradation persisted, making it possible to show that the CO net production rate decreased with increasing incubation time.

The resulting CO net production rate of 0.0038 – 0.0039 nmol L$^{-1}$ h$^{-1}$ (Fig. 7) is the first CO net production rate for an aquaculture effluent. It is extremely low compared to published CO production rates, which ranged from 1 to 106 nmol L$^{-1}$ h$^{-1}$ (Table 4). Notably, based on a comparison of the average CDOM absorbances (Table 5), our estimate of the average $S$ for the EPPO aquaculture effluent is higher than the values reported by Stedmon and Markager (2001) but within the range of or smaller than those reported by Højerslev et al. (2001), Nelson et al. (1998), Stedmon et al. (2000), and Kowalczuk et al. (2003). Additionally, our average value is nearly identical to that found in the North Atlantic (Babin et al. 2003). Although the CDOM absorption spectral slope differences seem to occur between the regions, the significant overall CO net production rate gaps we observed require further study and discussion. Under 48 hours of alternating day and night, the low CO net production indicates that microbial CO consumption was counteracting the CO photochemical production almost completely during incubation. Since there was no headspace in the incubation bottles, dissolved CO could not 'escape' consumption by being released into an overlying headspace, and thus microbial consumption efficiently reduced CO concentrations. Hence, our CO net production rates are most likely underestimated.

## 5 Conclusions

Dissolved CO surface concentrations were measured at 14 stations in the Ria Formosa Lagoon system in May 2021. The waters in the lagoon were found to be supersaturated with CO, and thus, Ria Formosa was a source of atmospheric CO. The distribution of dissolved CO concentrations correlated with FDOM, indicating that the photochemical production of CO was the major source of CO in Ria Formosa. Microbial CO consumption served as the primary sink for CO, while emissions to the atmosphere were of comparatively minor importance. Potential CO sources, such as phytoplankton and dark production,



appeared to be of minor importance. Overall, the CO cycling in the Ria Formosa Lagoon system aligned with observations
from other coastal regions. Therefore, our measurements support the idea that microbial CO consumption acts as a filter by
counteracting CO photoproduction and, in turn, effectively reducing subsequent CO emissions to the atmosphere. However,
the seasonality of CO cycling in the Ria Formosa Lagoon system is unknown. Furthermore, we showed that CO production
from the effluent waters of an aquaculture facility near the study area was low, which suggests that these aquaculture facilities
do not significantly contribute as a source of atmospheric CO.


Recent results show that increasing $NO_3^-$ concentrations can enhance the photoproduction of carbonyl sulfide (COS) (Li et al.,
2022). COS and CO photoproduction have a common intermediate in their photoproduction pathways, but the photoproduction
of COS and CO in natural waters is anticorrelated (Pos et al., 1998). The eutrophication of the Ria Formosa Lagoon leads to
high $NO_3^-$ concentrations with a pronounced seasonality (Domingues et al., 2017). Therefore, we contend that ongoing
eutrophication (i.e., increasing $NO_3^-$) might lead to lower CO photoproduction in the Ria Formosa Lagoon.

**Data availability**

All data have been submitted to PANGAEA database.
The monthly CO dry mole fractions from Mace Head and Terceira are available from the database of the NOAA Global
Monitoring Laboratory (Boulder, CO, USA) (https://gml.noaa.gov/dv/data/).

**Author contributions**

GL and RCOI designed the study and participated in the fieldwork; GL performed the measurements and wrote the manuscript;
DLAM, HWB, and RCOI contributed to the writing of the manuscript and discussion of the data.

**Competing interests**

The authors declare that they have no conflict of interest.

**Acknowledgements**

We express our gratitude to Dr. Ana Isabel Delfim dos Santos Alexandre and the staff at the Centro de Ciências do Mar
(CCMAR) of the University of Algarve in Faro for their invaluable assistance with the sampling and measurements during the
research campaign in Portugal. We also extend our appreciation to Dr. Ana Amaral (ASSEMBLE+ Liaison Officer) and Dr.
Rui Santos (Marine Plant Ecology Research Group) at CCMAR for generously providing laboratory space and access to



analytical instruments. Additionally, we would like to acknowledge the Instituto da Conservação da Natureza e das Florestas for granting us the necessary permit for fieldwork at Ria Formosa.

**Financial support**

Funding was provided by the ASSEMBLE Plus project, which has received funding from the European Commission's Horizon 2020 research and innovation program under grant agreement no. 730984 and the GEOMAR Helmholtz Centre for Ocean
Research Kiel. GL was supported by a CSC fellowship (grant no. 201908220144). RCOI was supported by a DAAD scholarship (PKZ 91726790).

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



**515  Figures**

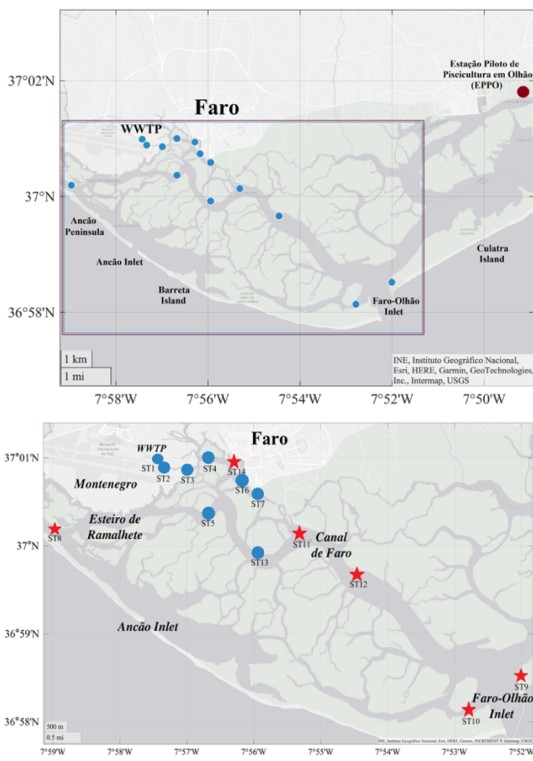

**Figure 1: Locations of sampling stations: Sites of water sampling (filled blue circles), atmospheric sampling (filled red stars) and**
**incubation experiments at the 'Estação Piloto de Piscicultura em Olhão (EPPO)' (filled red circles) are indicated.**

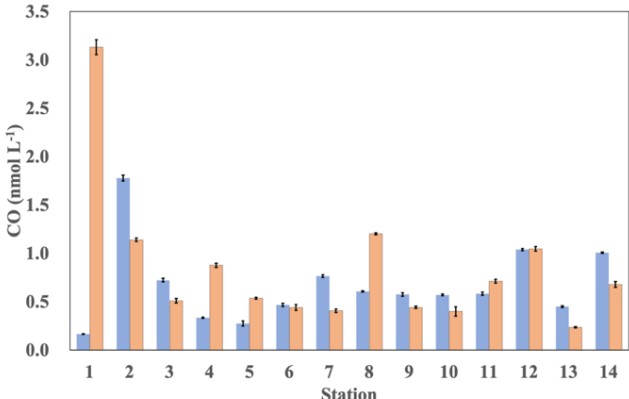

**Figure 2: Mean CO surface concentrations (± standard error estimate) at Stations 114 measured on 25 May 2021 (blue bars) and on**
**26 May 2021 (orange bars).**



**Figure 3: Surface distributions of temperature (°C), salinity, chlorophyll *a* (µg L$^{-1}$), pH, FDOM (QSU), CO concentrations (nmol L$^{-1}$), nitrate (µmol L$^{-1}$), and phosphate (µmol L$^{-1}$) in the Ria Formosa Lagoon in May 2021.**



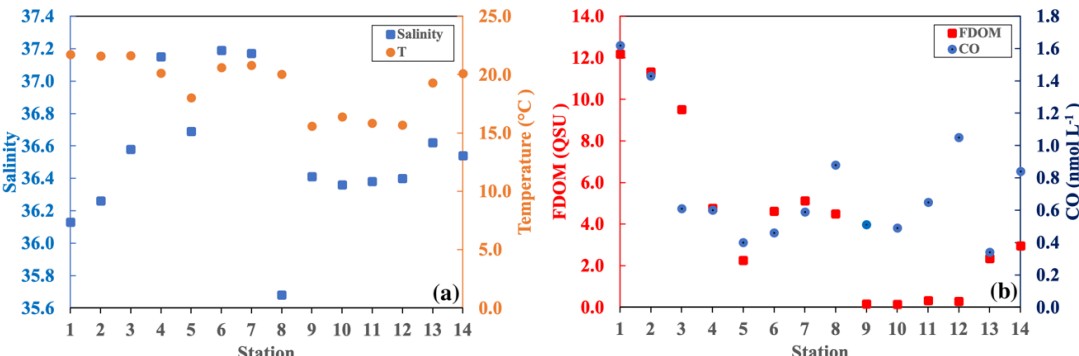

**Figure 4: Surface distributions of temperature (°C), salinity, chlorophyll $a$ (µg L$^{-1}$), pH, FDOM (QSU), CO concentrations (nmol L$^{-1}$), nitrate (µmol L$^{-1}$), and phosphate (µmol L$^{-1}$) in the Ria Formosa Lagoon in May 2021.**

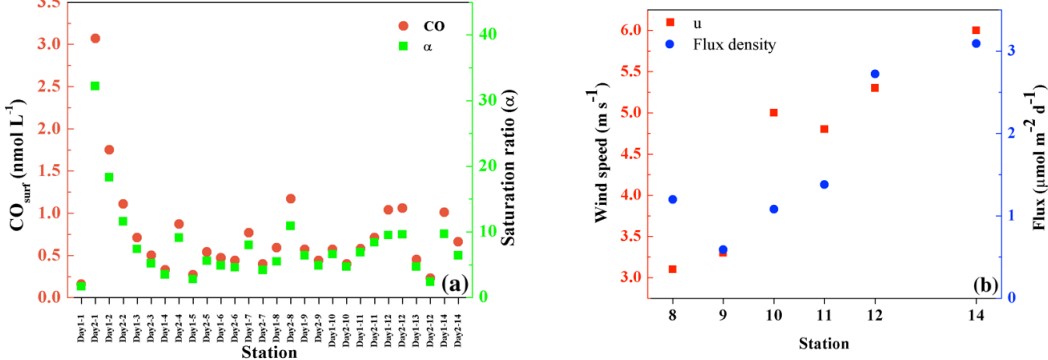

**Figure 5: Concentrations of dissolved CO (CO$_{surf}$; filled orange dots) and CO saturation ratios ($\alpha$; filled green squares) at all stations in the Ria Formosa Lagoon on 25 May (Day1) and 26 May 2021 (Day2) and (B) flux densities of CO (filled blue dots) and wind speeds (u$_{10}$; filled red squares) at Stations 8-12 and 14.**

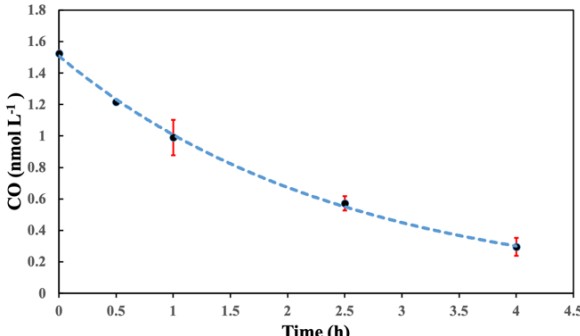



**Figure 6: CO concentrations (± standard error estimate) versus incubation time during the dark incubation experiment.**

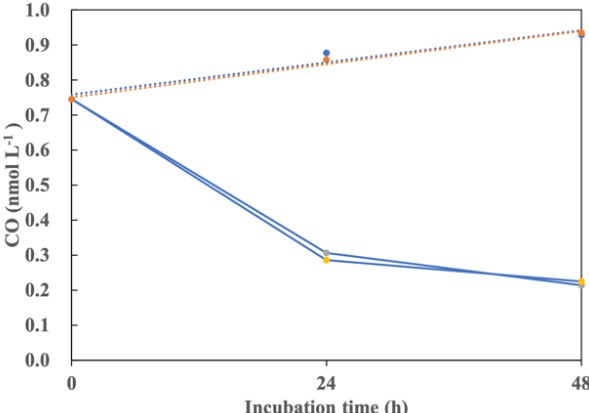

**Figure 7: Temporal variability of dissolved concentrations of CO in dark-group (solid line) and light-group (dashed line) incubations with water from the effluents of the aquaculture facility Estação Piloto de Piscicultura em Olhão (EPPO).**

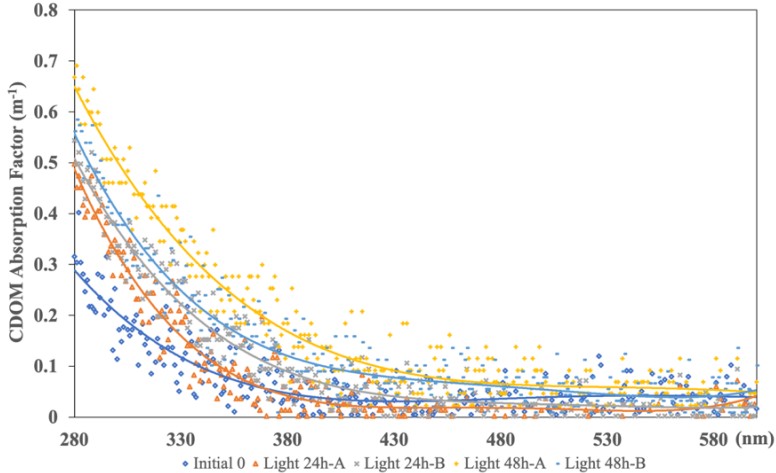

**Figure 8: The CDOM absorption spectra for different groups (Initial 0, Light-24h-A, Light-24h-B, Light-48h-A, Light-48h-B) of the aquaculture photo-incubations**





## Tables

**Table 1: Mean environmental parameters in Ria Formosa Lagoon (mean = the arithmetical average of the data from 25 and 26 May 2021).**

| Station | $CO_{surf}$ (nmol L$^{-1}$) | Chl-$a$ (µg L$^{-1}$) | FDOM (QSU) | Salinity | pH | T (°C) | $CO_{atm}$ (ppb) | $u_{10}$ (m s$^{-1}$) | $NH_4^+$ (µmol L$^{-1}$) | $PO_4^{3-}$ (µmol L$^{-1}$) | $NO_3^-$ (µmol L$^{-1}$) |
|---|---|---|---|---|---|---|---|---|---|---|---|
| 1 | 1.62 | 0.40 | 12.19 | 36.13 | 8.05 | 21.73 | — | — | 61.67 | 1.40 | 2.42 |
| 2 | 1.43 | 0.37 | 11.33 | 36.26 | 8.16 | 21.58 | — | — | 31.38 | 1.78 | 1.59 |
| 3 | 0.61 | 0.37 | 9.51 | 36.58 | 8.18 | 21.63 | — | — | ≤LOD | 0.23 | 0.85 |
| 4 | 0.60 | 0.23 | 4.76 | 37.15 | 8.18 | 20.14 | — | — | ≤LOD | 0.24 | 1.03 |
| 5 | 0.40 | 0.20 | 2.26 | 36.69 | 8.17 | 18.02 | — | — | ≤LOD | 0.30 | 0.94 |
| 6 | 0.46 | 0.27 | 4.62 | 37.19 | 8.22 | 20.61 | — | — | ≤LOD | 0.20 | 2.36 |
| 7 | 0.59 | 0.38 | 5.13 | 37.17 | 8.21 | 20.78 | — | — | ≤LOD | 0.29 | 2.16 |
| 8 | 0.88 | 0.50 | 4.50 | 35.68 | 7.97 | 20.04 | 126 | 3.1 | ≤LOD | 0.30 | 0.35 |
| 9 | 0.51 | 0.04 | 0.16 | 36.41 | 8.12 | 15.57 | 98 | 3.3 | ≤LOD | 0.36 | 2.19 |
| 10 | 0.49 | 0.02 | 0.14 | 36.36 | 8.11 | 16.37 | 95 | 5.0 | ≤LOD | 0.50 | 3.31 |
| 11 | 0.65 | 0.05 | 0.32 | 36.38 | 8.12 | 15.85 | 93 | 4.8 | ≤LOD | ≤LOD | 1.49 |
| 12 | 1.05 | 0.04 | 0.29 | 36.40 | 8.10 | 15.67 | 121 | 5.2 | ≤LOD | 0.80 | 3.19 |
| 13 | 0.34 | 0.14 | 2.34 | 36.62 | 8.18 | 19.28 | — | — | ≤LOD | 0.54 | 0.70 |
| 14 | 0.84 | 0.23 | 2.95 | 36.54 | 8.18 | 20.08 | 124 | 6.1 | ≤LOD | 0.54 | 1.00 |

560

**Table 2: Major sources and sinks (in µmol m$^{-2}$ d$^{-1}$) of CO in seawater in the Ria Formosa Lagoon in May 2021.**

| *Sources* | |
|---|---|
| Photoproduction | 8.78 |
| Dark production | $3.66 \times 10^{-5}$ |
| Production by phytoplankton | $5.38 \times 10^{-5}$ |
| *Sinks* | |
| Sea-to-air flux density | 1.53 |
| Microbial consumption | 7.25 |

565



**Table 3: List of CDOM absorption spectra ($\underline{S}_{280-295}$ and $\underline{S}_{350-400}$) and $\underline{S}_R$ for different groups (Initial 0, Light-24h-A, Light-24h-B, Light-48h-A, Light-48h-B) of the aquaculture photo-incubations.**

|  | $S_{280-295}$ | $S_{350-400}$ | $S_R$ |
|---|---|---|---|
| Initial 0 | 0.02 | 0.036 | 0.56 |
| Light 24h-A | 0.013 | 0.031 | 0.42 |
| Light 24h-B | 0.017 | 0.027 | 0.63 |
| Light 48h-A | 0.019 | 0.019 | 1 |
| Light 48h-B | 0.017 | 0.015 | 1.13 |

**Table 4: List of CO production rates from irradiation experiments with seawater samples from coastal and open ocean sites.**

|  | CO production, nmol L$^{-1}$ h$^{-1}$ | Reference |
|---|---|---|
| Eastern Caribbean Sea | 18.5 – 25.5[a] | Jones and Amador, 1993 |
| Gulf of Paria, off Venezuela | 32.5 – 36.6[a] | Jones and Amador, 1993 |
| Orinoco River plume, tropical NW Atlantic | 19.1 – 88.3[a] | Jones and Amador, 1993 |
| Pettaquamscutt River estuary, NW Atlantic | 1 – 20[b] | Schmidt and Heikes, 2014 |
| Vineyard Sound, NW Atlantic | 0.83[c] / 2.35[d] | Xie and Zafiriou, 2009 |
| Bermuda Atlantic Time-Series Study Site | 0.034[c] / 0.14[d] | Xie and Zafiriou, 2009 |
| Pointe-au-Père, St. Lawrence Estuary | 0.31[c] / 2.79[d] | Xie and Zafiriou, 2009 |
| Biscayne Bay, NW Atlantic | 28.1[e] | Pos et al., 1998 |
| Intracoastal Waterway, off Florida, NW Atlantic | 110[f] | Valentine and Zepp, 1993 |
| Live Oak, off Florida, Gulf of Mexico | 90[f] | Valentine and Zepp, 1993 |
| Gulf of Mexico/Sargasso Sea | 11 – 13[f] | Zuo and Jones, 1995 |
| Gulf Stream off Florida, NW Atlantic | 12 – 106[f] | Zuo and Jones, 1995 |
| Scheldt Estuary, North Sea | 1 – 10.6[g] | Law et al., 2002 |

[a] normalized to 1000 Wh m$^{-2}$ irradiance, irradiation with natural sunlight

[b] calculated as $K_{bio}$ * max. CO concentration during the day

[c] production from particles, irradiation with solar simulator

[d] production from CDOM, irradiation with solar simulator

[e] filtered samples, irradiation with solar simulator

[f] irradiation with solar simulator

[g] irradiation with natural sunlight



**Table 5: List of spectral slope values for marine samples reported in the literature with spectral range.**

| Location | S (nm$^{-1}$) | Wavelength range | Reference |
|---|---|---|---|
| Kattegat–Skagerrak | 0.0234 ± 0.0036 | 250–450 | Højerslev et al. (2001) |
| Bermuda | 0.0235 | 280–350 | Nelson et al. (1998) |
| Danish fjords and nearby coastal waters | 0.0194 ± 0.0032 | 300–650 | Stedmon et al. (2000) |
| Greenland Sea, Jun | 0.01651 ± 0.00352 | 300–650 | Stedmon and Markager (2001) |
| Baltic, coastal | 0.020 ± 0.003 | 350–var | Kowalczuk et al. (2003) |
| North Atlantic | 0.0176 ± 0.002 | 350–var | Babin et al. (2003) |
| 'Estação Piloto de Piscicultura em Olhão' effluent | 0.0176 ± 0.004 | 280–330 | This study |

590