# Peer review of "Carbon monoxide cycling in the Ria Formosa Lagoon (southern Portugal) during summer 2021"

_EGUsphere, 2023_

## Author Comment (AC1)

We sincerely thank Referee 1 for the valuable comments we used to improve the quality of this manuscript. According to these suggestions, we have supplemented several references and corrected some mistakes in our previous draft. These are listed below.

However, I have a major concern on the quality of the data and thus the resulting conclusions. My concern primarily arises from the extremely sparse sampling resolutions: only one time point (late afternoon and high tide) on two days for CO concentration sampling and only one station (Sta. 7) for CO consumption sampling. It is well known that surface water CO concentration changes substantially over diel time scales mainly imposed by daytime photoproduction and bacterial consumption throughout day and night. This diel cycle can be further complicated by tidal variation in coastal areas. It is also well recognized that microbial CO consumption presents large spatiotemporal variations in coastal regions. The data obtained from such poor sampling resolutions can only capture a slim part of the diel cycle and thus cannot support the conclusions reported. For example, if we use the $K_{bio}$ determined at Sta. 7 (0.40 h$^{-1}$) to estimate the surface water CO concentration in early morning (before dawn, assuming 7 hours of nighttime) at this same locality, we get 0.04 nM, which is below the CO concentration at equilibrium with air (~0.1 nM). Then, the late-afternoon source turns into a nighttime sink. This situation may apply to many other stations, because the reported surface water CO concentrations were mostly quite low.

Reply: Thank you for pointing out this issue. It would have been better if more sampling sites with a higher frequency were sampled to confirm the conclusion of our study. Nevertheless, we think our spatial resolution is adequate to capture the environmental gradients of relevance for CO cycling in the study area. By collecting data at the same specific locations in two days, we can ensure that the data was obtained under similar environmental conditions in the area of influence of low human activities, reducing uncertainties caused by site variations. In the WWTP effluent areas, samples were collected at the same stations at two time periods, both when no effluent was discharged and when effluent was discharged, to better determine its influence on CO. Besides, the sampling time was limited by the tide and the logistical constrain; it was imperative to consider minimizing the interval between sampling and measurement to minimize CO sample uptake within the vial and microbial consumption processes.

The consumption sampling station was located in a representative area for the combined effects of tidal effects and terrestrial contributions, and therefore we considered it to be a sampling site that may feature a good approximation of the rate of microbial consumption in this region.

The second important factor that led to me casting doubt on the quality of the data is the huge correction factor (3.12) for CO concentrations. There is no explanation for what caused the technical problem leading to this correction. Any time- and concentration-dependence of the correction factor? The uncertainty of your atmospheric CO measurement was 14.2%, which alone leads to significant variations in the correction factor. This variability can be larger if the uncertainties of the atmospheric CO measurements at Mace Head and Terceira are taken into account. Please present an uncertainty analysis of this correction.

Reply: Please see the reply to the specific comment on this.

In addition, there are errors in the calculation of CO concentrations (eq. 1, see specific comments on this).

Reply: Please see the reply to the specific comment on this.

The CO budget (eq. 11) is unreliable because the poor sampling resolutions invalidate the steady-state assumption. Moreover, the budgeting misses a potentially important source (terrestrial input including the WWTP effluent) and a potentially important sink (output to the outside sea).

Reply: We disagree. The low resolution might impact the robustness of the results but not necessarily the steady-state assumption. In this study, all the sources and sinks mentioned were considered to be a sort of vertical transport process. The lateral terrestrial input (including WWTP effluent) with high FDOM and nutrients contributed to vertical processes like CO photodegradation, biological production, and microbial consumption. Budgets for the lateral transport processes are indeed very helpful to a complete understanding of their area. However, they were not part of the study presented here.

The interpretation of the data from the incubations of the aquaculture is superficial and often inaccurate or incorrect (see specific comments). In fact, the data allows you to derive the gross production in the "light" treatment.

Reply: Please see the reply to the specific comment on this.

Some suggestions:

Make sure that the calculations of and corrections on CO concentrations are correct.

Base your conclusions on the data. Do not over-extrapolate temporally and/or spatially.

Reply: Many thanks for your suggestions, and we have replied to each specific comment individually.

**Specific comments:**

Introduction: Put your study in a broad context: why do you choose a human-impacted lagoon system?

Reply: According to the comment, we integrated the last paragraph of the introduction:

Ria Formosa is a major lagoon system in southern Portugal (36°58' N, 8°02' W to 37°03' N, 7°32' W) (Newton and Mudge, 2003; Cravo et al., 2014). It is a mesotidal coastal lagoon with an average water depth of ~2 m. Water temperatures typically range from about 12 °C in the winter to about 27 °C in the summer (Newton and Mudge, 2003). Salinities range from 13 to 36.5 (Newton and Mudge, 2003). The lagoon is vertically well-mixed due to limited freshwater inputs and the predominance of a tidal-forced circulation pattern (Cravo et al., 2014). Being a typical coastal lagoon system, the water exchange with the adjacent shelf is restricted (Tett et al., 2003). In addition

to nutrient inputs from aquaculture facilities, the Ria Formosa Lagoon receives significant nutrient inputs from agricultural runoff and sewage, particularly from the cities of Faro, Olhão, and Tavira (Newton and Mudge, 2005). A part of the Ria Formosa Lagoon system is a natural park established in 1987 (Aníbal et al., 2019). Ria Formosa also plays a vital role in the region's tourism industry, as well as supporting a fishery and aquaculture industry of national significance. Due to its restricted (estuarine-type) circulation and high nutrient and organic matter inputs, the Ria Formosa Lagoon system could be a potential source of atmospheric CO. The unique setting at the Ria Formosa lagoon system provides an excellent opportunity to investigate the dynamics of carbon monoxide in an anthropogenically-influenced coastal system.

Additionally, eutrophication and algal bloom often occur in aquaculture ponds in Ria Formosa, which should be a non-negligible component of the RF-CO cycle process. How these environmental factors will affect CO production and emissions from the lagoon system (accounting for approximately 13% of the coastal areas worldwide) is unknown due to limited measurements and knowledge gaps regarding its sources and sinks. In this study, we present the first in situ observations of CO in the region, which we combined with incubation experiments in waters directly influenced by anthropogenic activities. The main objectives of this study were to i) elucidate the CO distribution and sea–air flux densities in the coastal lagoon system, ii) estimate the different CO sources and sinks in the lagoon system during summer, and iii) determine the potential impact of aquaculture activities on CO cycling in this region.

L32: Ossola et al. (2022) used model compounds not natural CDOM and their experiments were conducted at pH 5.8-6.5, conditions atypical of coastal and open oceans. Replace it with a more appropriate reference.

Reply: Ossola et al. (2022) was replaced with Stubbins et al. (2006).

Integrate "Study site description" into the last paragraph of the "Introduction". Elaborate more about the human activities (including aquaculture) in the region.

Reply: See our first reply to the specific comment, and we added more details about human activities (including aquaculture) in the region: In addition to nutrient inputs from aquaculture facilities, the Ria Formosa Lagoon receives significant nutrient inputs from agricultural runoff and sewage, particularly from the cities of Faro, Olhão, and Tavira (Newton and Mudge, 2005). A part of the Ria Formosa Lagoon system is a natural park established in 1987 (Aníbal et al., 2019). Ria Formosa also plays a vital role in the region's tourism industry, as well as supporting a fishery and aquaculture industry of national significance.

L63-64: Give more details about how seawater was transferred from the Niskin bottles to the quartz glass bottles (e.g., the material of the tube used for sample transfer; if the bottles were overflowed; if sunlight was avoided during the transfer; what caps were used to close the bottles, etc.).

Reply: Here we added more details about the sample transfer process: Bubble-free water samples were collected as triplicates for the determination of dissolved CO and were transferred to 100 mL quartz glass bottles by using Tygon® tubing to avoid contamination by silicone rubber (Xie et al., 2002) from the Niskin bottle, taking care to keep the tube at the bottom of the vail and control the

speed of water flow to avoid bubbles and swirls. Our samples were immediately sealed with rubber stoppers and aluminum caps, stored between 0 and 4 °C in the dark to suppress further CO photoproduction and uptake within the vial at ambient conditions (Day and Faloona, 2009), and transported within 1.5 hours to the Centro de Ciências do Mar (CCMAR) for analysis.

L72: Samples for dark incubation were collected only on one day? Then was it May 25 or 26?

Reply: The samples for the dark incubation (microbial CO consumption experiment) were collected only on 26 May. For more clarity, we adapted the text as follows: Water duplicate samples from Station 7 (the representative area for the combined effects of tidal effects and terrestrial contributions) for determining CO consumption on May 26th were introduced into 100 mL quartz glass bottles from the Niskin bottle and then covered with aluminum foil and stored in a cooling box (0~4 °C) after sampling.

L73: How many replicate incubations?

Reply: We did the microbial CO consumption experiment once for duplicate water samples on May 26th. For more clarity, we adapted the text as follows: Water duplicate samples from Station 7 (the representative area for the combined effects of tidal effects and terrestrial contributions) for determining CO consumption on May 26th were introduced into 100 mL quartz glass bottles from the Niskin bottle and then covered with aluminum foil and stored in a cooling box (0~4 °C) after sampling.

L74: "cooling box"—what was the temperature inside the box, well below the in situ temperature? If so, could significantly affect bacterial activity.

Reply: The temperature of the cooling box was between 0 and 4 °C. For more clarity, we adapted the text as follows: From Fig. S1, it is clear that after sampling and transporting the samples to the CCMAR for analysis within 0.5 h, the CO loss is less than 5%.

L75-78: Was temperature stable during the incubation? At what temperature? At or close to in situ temperature?

Reply: During the incubation experiment, all samples were stored at an in-situ temperature maintained at ± 1 °C. For more clarity, we adapted the text as follows: All samples for the consumption experiment were transported back to the laboratory within half an hour and stored at an in-situ temperature maintained at ± 1 °C in the lab.

CO measurements: What was the precision and detection limit of the method.

Reply: The detection limit of our measurements of dissolved CO was 10 ppb, and the analytical precision for the method was ± 0.011 nmol L$^{-1}$. We added an explanation to the revised manuscript as follows: CO in the seawater and atmosphere was detected using a CO Gas Analyzer (ta3000R; Ametek, USA) with a lower detection limit of 10 ppb, and the analytical precision for the method was ±0.011 nmol L$^{-1}$.

L81: Did the 113.9 ppb standard cover the range of CO concentrations in the headspace of your samples? If not, what was the linear response range of the CO analyzer and was the range large enough to cover your samples' headspace concentrations?

Reply: The system was calibrated with the CO standard every half hour, or before and after related sample blocks, by replicating injections. For our datasets, it is convenient to correct data by interpolating standard peak areas over time periods between calibrations, like other studies using a similar method (e.g., Xie et al. 2002). Besides, the linear response range of the analyzer is 0~3 ppm CO. The standard gas was chosen as it lies in the expected range of the CO mole fraction equilibrated with the study area.

L85-89: Provide more details about how the headspace was created and how the headspace gas was subsampled (i.e., creation of headspace: how water was removed from and how gas was injected into the bottle? Subsampling the headspace gas: how the headspace gas was sampled? During both the creation and subsampling of the headspace gas, how the headspace was maintained at the atmospheric pressure?)

Reply: A syringe with a short needle was used as a conduit to introduce CO-free gas (12 mL) into a vial. The inflow of CO-free gas expels the water sample through the longer needle, creating a headspace within the vial (Fig. 1). The vial was equilibrated at room temperature and atmospheric pressure for 5 minutes at 120 rpm on a KS-90 shaking table (Edmund Bühler, Germany), followed by an additional equilibration period of 3 min before analysis. Then, using a gas-tight Luer-lock syringe, we extracted the equilibrium gas as the gaseous subsample of the headspace with the CO analyzer. This information was added to the revised manuscript.

[Figure]

Figure 1. A sketch of the headspace introduction apparatus.

Equations 1-5: List units for all parameters. Is Vw the volume before making the headspace or after making the headspace? Beta in Wiesenburg and Guinasso (1979) is in mL gas (mL $H_2O)^{-1}$ $atm^{-1}$. Moreover, the units of R are atm L $mol^{-1}$ $K^{-1}$. How did you get the units of nmol $L^{-1}$ for $CO_{surf}$ (L94)?

Reply: We apologize for the confusion we caused. Units for all parameters have been added to the manuscript (see e.g., $\beta$ (mL CO (mL $H_2O)^{-1}$ $atm^{-1}$)). $V_w$ is the volume of the water sample after making the headspace (88 mL), and $V_a$ is the volume of the headspace air (12 mL). In the second term of equation 1, the unit is in mol $L^{-1}$. According to Equation 7 and Table 4 (Wiesenburg and Guinasso, 1979), constants for the calculation of solubilities in nmol/L (CO) from moist air at 1 atm total pressure (first term of equation 1). Then, we could get the $CO_{surf}$ by the unit of nmol $L^{-1}$.

Equation 1: The equation seems incorrect. The first term should be multiplied "P" and "Vw (volume of remaining water after the creation of headspace) then divided by the volume of the water before the creation of the headspace.

Reply: We disagree. We simplified Eq. 1: the first term should be multiplied by 'P' and '$V_w$' indeed, then divided by '$V_w$', not the volume of the water before the creation of the headspace. 'P' was 1 atm.

L104: "in equilibrium with the headspace". Note COeq is the dissolved CO concentration in the water remaining after the creation of the headspace.

Reply: We see that there has been a misunderstanding. We corrected and revised the equation and text as follows: The dissolved CO concentration in equilibrium with surface water ($CO_{eq}$ in mol $L^{-1}$) was calculated using Eq 6 (Wiesenburg and Guinasso, 1979), x'$_{atm}$ is the atmospheric CO dry mole fraction:

$$CO_{eq} = x'_{atm} \times \beta \qquad (6)$$

Equation 4 is the first term of equation 1. If Ceq here refers to the CO concentration in surface water in equilibrium with the atmosphere (see equation 6), then x' here refers to the dry mole fraction of CO in the air samples. Do not mix this x' with that in equation 1.

Reply: We are sorry for the confusion. We have corrected and revised the equations:

$$CO_{eq} = x'_{atm} \times \beta$$

L107: Please explain what was the technical problem? The correction fact f = 3.12. This was huge!

Reply: We acknowledged that the issue stems from a malfunction in the calibration system – only errors in the calibration algorithm, not issues with the detector or lamp: The CO analyzer did not report any error during the warm-up progress including self-test after power-on, indicating that the detector, carrier gas pressure, and lamp were working normally. We repeated calibrations over time, using the same standard gases, to ensure that the detector consistently responds to the point calibration. The results were consistent and reproducible. However, the values obtained from the analyzer during the calibration were only about one-third of those given for the standard gases.

Further, we compared the average atmospheric CO mole fraction for the same month (May 2021) from atmospheric monitoring stations in Mace Head and Teixeira. A correction factor f (3.12) was determined. This was the fundamental reason for using the correction factor f for the CO analyzer measurements.

L111-113: "nearly uniform atmospheric background CO mole fractions". This is usually true over open oceans but not near the coast. The relative standard deviation of your measurements (5/35.1=14.2%) indicates this.

Reply: Thank you for pointing out this issue. The data obtained from the nearshore atmosphere were variable. But from the aspect of the monthly mean atmospheric background CO mole fractions across the Northwestern Atlantic Ocean during our measurement period were nearly uniform.

Equation 6. See comment on equation 4.

Reply: We have corrected and revised the equations.

L124-129: Did you measure u2?

Reply: Yes, instantaneous wind speeds at 2 m above sea level were measured onboard during atmospheric sampling using a vane anemometer (Testo, Germany). This information will be added to the revised manuscript.

Equation 7: Flux densities were calculated using spot windspeeds? If not, how windspeeds were processed? (if averaged, give the relevant spatial and temporal scales).

Reply: Flux densities were calculated using spot windspeeds (see Table 1).

Figure 3: Temperature, pH, and nutrients are shown in Figure 3 but not cited in the text.

Reply: Temperature, pH, and nutrients were the important basis for distinguishing the WWTP thermal effluent plume zone. We discuss them in lines 180 to 197.

Figure 2: What is "Stations 114" in the figure caption?

Reply: Thanks for spotting this mistake. We have corrected the caption of Fig.2. 'Figure 2: Mean CO surface concentrations (± standard error estimate) at Stations 1–14 measured on 25 May 2021 (blue bars) and on 26 May 2021 (orange bars).'

L180: Justify using one station (Sta. 18) to represent a region (Praia de Faro zone).

Reply: In the paper, we intended to point out that the chosen station 8 is close to the "Praia de Faro" zone. We did not use one station (Sta. 8) to represent a region (Praia de Faro zone). Notably, during the high tide, the water of the Faro channel from the region of Montenegro through Esteiro Largo and to the West region through the Ramalhete. Station 8 was a very representative sampling location.

L188: CO concentration at Sta. 13 was among the lowest (Figures 2 and 4).

Reply: Thanks for pointing this out. We have corrected station 13 to station 14.

L188-189: "biological CO production". This is purely speculative. FDOM at Sta. was high as well.

Reply: We agree that this idea is very speculative. However, we would like to keep the small paragraph as it seems justified by the results presented by Gros et al. (2009) and McLeod et al. (2021). The production and distribution of CO were affected by multiple factors; CO was also produced by phytoplankton, albeit to a minor extent.

L189-191: "WWTP thermal effluent plume". What was your purpose of mentioning this plume?

Reply: Here, we would like to illustrate that at station 8, CO concentrations were also affected by anthropogenic (e.g., high FDOM) influence.

L196-197: n = 14 for the relationship between CO and phosphate, but n = 28 for the relationship between CO and FDOM. Why? Any implication for the significant correlation between CO and phosphate?

Reply: Because of logistical constraints in the number of phosphate samples we could take. There was no valid evidence for the phosphate to directly affect the photodegradation of CDOM. Still, it was also an important factor affecting the distribution of CO at the lagoon system. More studies are needed to explain the positive correlation between these two better.

L201-202: "The lagoon was a source of atmospheric CO in May 2021". This is a premature conclusion since you only sampled on two days, each at a specific time point and tidal level (see general comment). Better to say "The lagoon was source of atmospheric CO at the time of sampling".

Reply: Thank you for pointing out this issue, and we adapted the text as follows: This indicates that the lagoon was a source of atmospheric CO at the time of sampling, which is in line with the observations from other coastal waters.

Figures 3, 4, and 5(b) and Table 1: Samples were collected on two days. Are the data shown in these figures the average of the two samples? If averaged, what are the variabilities of the two sampling times? Justify the use of average values.

Reply: Figures 3, 4, and Table 1 were the averages of two days' data, and Fig.5(b) was the flux densities of CO and wind speeds at Stations 8-12 and 14 on day two (we have corrected the caption). One aspect of our averaging was because some of the data were obtained from CCMAR (nutrients). More important was the point that CO concentrations in this study area were strongly affected by environmental conditions with anthropogenic activities, especially the WWTP. Thus, averaging the samples from the two days with and without WWTP emissions can better evaluate the CO distribution and potential impact of other activities on CO cycling in this region.

Figure 8: What do you mean by "CDOM absorption factor"?

Reply: We have replaced the 'CDOM absorption factor' with 'CDOM absorption coefficient'.

L266-267: Why CO concentration did not decrease during night? Hard to understand. Bacteria were killed by solar UV during daytime?

Reply: There seems to be a misunderstanding. Microbial consumption was always persisting, both during the day and at night. The experiments with sampling from the incubation bottles were conducted at time points 0, 24, and 48 h. In this 48-hour incubation experiment, the total photogeneration of CO was greater than microbial consumption, so it reflected continuous growth.

L267-269: In fact, the dark incubation roughly followed an exponential decay trend. No evidence suggests that a steady state reached at the end of the incubation.

Reply: To clarify our statement, we modified it. It reads now, 'This indicates that microbial CO consumption, probably restricted by CO dark production, led to a lower consumption state at the end of the dark incubation.'

L284-289: Much of the discussion is purely speculative.

Reply: We are sorry, but we disagree. We would like to keep the small paragraph as it seems to be justified by the results presented in Ossola et al. (2022) and Xu et al. (2023):

The ratio of the spectral slope ($S_R$) of the shorter waveband (280–295 nm) to that of the longer waveband (350–400 nm) was used to identify the origin of the CDOM: the smaller the $S_R$, the higher the proportion of terrestrial CDOM, and vice versa (Helms et al., 2013; John et al., 2009; Zhu et al., 2017). Besides, Xu et al. (2023) pointed out that there was a significant negative correlation between the apparent efficiency of CO photoproduction and $S_R$. Ossola et al. (2022) demonstrated for the first time a precise mechanism for CO production: the aromatic methoxy groups can produce CO. Summarizing, using $S_R$, the CO photoproduction capacity can be more accurately evaluated.

L291-293: Such a comparison does not mean much because you measured net production while the others reported photoproduction.

Reply: Thank you for pointing to this issue. We changed the comparison about the photoproduction in different areas of the manuscript.

**Technical corrections:**

L16: remove "of the Ria Formosa Lagoon".

Reply: Done.

L21: replace "it" with "this reaction".

Reply: Done.

L27: remove "may".

Reply: Done.

L28: "early" à "earlier"; "indicate"à"reported".

Reply: Done.

L125: at 10 m.

Reply: Done.

Equation 9: define u2.

Reply: Done.

L161: in the Ria…

Reply: Done.

L175: Show the Ramalhete and Faro–Olhão inlets in the maps of Figure 3.

Reply: Done.

L176: replace first "." with ",".

Reply: Done.

Figure 4: Caption is incorrect (now the same as that of Figure 3). Chl-a is missing.

Reply: Many thanks. We have corrected the caption: 'Figure 4: (A) mean salinity (filled light blue squares) vs. mean temperature (filled orange dots) and (B) mean concentration of dissolved CO (filled blue dots) and mean FDOM (filled red squares) at all stations in the Ria Formosa Lagoon on 25 May (Day1) and 26 May 2021 (Day2).'

L197: add "," after "r = 0.860".

Reply: Done.

Figure 8: Absence of symbol legends. What do you mean by "CDOM absorption factor"?

Reply: Done, and we have replaced the 'CDOM absorption factor' with 'CDOM absorption coefficient'.

**References**

Aníbal, J., Gomes, A., Mendes, I., and Moura, D.: Ria Formosa: challenges of a coastal lagoon in a changing environment, http://hdl.handle.net/10400.1/12475, 2019.

Cravo, A., Cardeira, S., Pereira, C., Rosa, M., Alcântara, P., Madureira, M., Rita, F., Luis J. and Jacob, J.: Exchanges of nutrients and chlorophyll a through two inlets of Ria Formosa, South of Portugal, during coastal upwelling events, Journal of Sea Research, 93, 63-74, https://doi.org/10.1016/j.seares.2014.04.004, 2014.

Day, D. A. and Faloona, I.: Carbon monoxide and chromophoric dissolved organic matter cycles in the shelf waters of the northern California upwelling system, Journal of Geophysical Research: Oceans, 114(C1), https://doi.org/10.1029/2007JC004590, 2009.

Gros, V., Peeken, I., Bluhm, K., Zöllner, E., Sarda-Esteve, R., and Bonsang, B.: Carbon monoxide emissions by phytoplankton: evidence from laboratory experiments, Environmental Chemistry, 6(5), 369-379, https://doi.org/10.1071/EN09020, 2009.

Helms, J. R., Stubbins, A., Perdue, E. M., Green, N. W., Chen, H., and Mopper, K.: Photochemical bleaching of oceanic dissolved organic matter and its effect on absorption spectral slope and fluorescence, Marine Chemistry, 155, 81–91, https://doi.org/10.1016/j. marchem.2013.05.015, 2013.

John, R. H., Stubbins, A., Ritchie, J. D., Minor, E. C., Kieber, D. J., and Mopper, K.: Absorption spectral slopes and slope ratios as indicators of molecular weight, source, and photobleaching of chromophoric dissolved organic matter. Limnology & Oceanography, 54(3), 1023. https://doi.org/10.4319/lo.2009.54.3.1023, 2009.

McLeod, A. R., Brand, T., Campbell, C. N., Davidson, K., and Hatton, A. D.: Ultraviolet Radiation Drives Emission of Climate-Relevant Gases from Marine Phytoplankton, Journal of Geophysical Research: Biogeosciences, 126(9), e2021JG006345, https://doi.org/10.1029/2021JG006345, 2021.

Newton, A., and Mudge, S. M.: Temperature and salinity regimes in a shallow, mesotidal lagoon, the Ria Formosa, Portugal. Estuarine, Coastal and Shelf Science, 57(1-2), 73-85, https://doi.org/10.1016/S0272-7714(02)00332-3, 2003.

Newton, A., and Mudge, S. M.: Lagoon-sea exchanges, nutrient dynamics and water quality management of the Ria Formosa (Portugal). Estuarine, Coastal and Shelf Science, 62(3), 405-414, https://doi.org/10.1016/j.ecss.2004.09.005, 2005.

Ossola, R., Gruseck, R., Houska, J., Manfrin, A., Vallieres, M., and McNeill, K.: Photochemical production of carbon monoxide from dissolved organic matter: Role of lignin methoxyarene functional groups. Environmental Science and Technology, 56(18), 13449–13460. https://doi.org/10.1021/acs.est.2c03762, 2022.

Stubbins, A., Uher, G., Law, C. S., Mopper, K., Robinson, C., and Upstill-Goddard, R. C.: Open-ocean

carbon monoxide photoproduction, Deep Sea Research Part II: Topical Studies in Oceanography, 53(14-16), 1695-1705, https://doi.org/10.1016/j.dsr2.2006.05.011, 2006.

Tett, P., Gilpin, L., Svendsen, H., Erlandsson, C. P., Larsson, U., Kratzer, S., ... and Scory, S.: Eutrophication and some European waters of restricted exchange, Continental Shelf Research, 23(17-19), 1635-1671, https://doi.org/10.1016/j.csr.2003.06.013, 2003.

Wiesenburg, D. A., and Guinasso Jr, N. L.: Equilibrium solubilities of methane, carbon monoxide, and hydrogen in water and seawater, Journal of Chemical and Engineering Data, 24(4), 356-360, 1979.

Xie, H., Andrews, S. S., Martin, W. R., Miller, J., Ziolkowski, L., Taylor, C. D., and Zafiriou, O. C.: Validated methods for sampling and headspace analysis of carbon monoxide in seawater, Marine Chemistry, 77(2-3), 93-108, https://doi.org/10.1016/S0304-4203(01)00065-2, 2002.

Xu, G. B., Xu, F., Ji, X., Zhang, J., Yan, S. B., Mao, S. H., and Yang, G. P.: Carbon monoxide cycling in the Eastern Indian Ocean. Journal of Geophysical Research: Oceans, e2022JC019411, https://doi.org/10.1029/2022JC019411, 2023.

Zhu, W. Z., Yang, G. P., and Zhang, H. H.: Photochemical behavior of dissolved and colloidal organic matter in estuarine and oceanic waters. Science of the Total Environment, 607–608, 214–224. https://doi.org/10.1016/j.scitotenv.2017.06.163, 2017.

---

## Author Comment (AC2)

We are thankful for the thoughtful and constructive comments by Dr. Hong-Hai Zhang. We added corrections/additions to the manuscript (in blue font). Our point-by-point responses are as follows:

R1: Page 4 Lines 107: The author mentioned in the manuscript that because of a technical problem with the calibration of the CO analyzer, data was corrected by the correction factor. It is not clear for me that the correction factor is for atmospheric samples or for all samples?

Reply: We apologize for any confusion caused. We acknowledged that the issue stems from a malfunction in the calibration system. There seemed to be errors with the calibration algorithm: The CO analyzer did not report any error during the warm-up progress including self-test after power-on, indicating that the detector, carrier gas pressure, and lamp were working normally. We repeated calibrations over time, using the same standard gases, to ensure that the detector consistently responds to the point calibration. The results were consistent and reproducible. However, the values obtained from the analyzer during the calibration were only about one-third of those given for the standard gases. Further, we compared the average atmospheric CO mole fraction for the same month (May 2021) from atmospheric monitoring stations in Mace Head and Teixeira. A correction factor $f$ (3.12) was determined. This was the fundamental reason for using the correction factor $f$ for the CO analyzer measurements. Fieldwork and incubation samples were all corrected by the factor. The corrected data can be viewed in Table 1 or in the supplement.

R2: Page 9 Lines 263-264: as mentioned in the previous paragraphs 'with the assumption of a steady state, the sum of the CO sources and sinks is equal to zero'. It should not be mentioned here again in the form of a conclusion. It is recommended to modify or delete.

Reply: We will delete 'with the assumption of a steady state, the sum of the CO sources and sinks are equal to zero'.

R3: For the microbial CO consumption experiment, should the influence of dark (thermodynamic) production on CO be considered?

Reply: We think the estimates of dark (thermodynamic) production in the Ria Formosa Lagoon (in Table 2) were several orders of magnitude below, which was negligible compared to microbial CO consumption.

R4: should the sampling density be increased, especially between 0 and 24 hours? And the author mentioned that CO net production rate for Olhão aquaculture effluent is extremely low, microbial CO consumption was counteracting the CO photochemical production almost completely. Would the author consider using some testing methods to analyze the consumption of microorganisms to confirm the experimental results?

Reply: Thank you for pointing to this issue. The suggested experiments would indeed be beneficial to confirm the conclusion of our study. However, they were not part of the study presented here.

R5: The conclusion should not contain too many references.

Reply: We think that citing three references in the Conclusion section is not a case of 'overciting'. Moreover, the cited references are all necessary to justify our conclusions.

R6: Minor comments for the figures:
The quality of the figure should be improved (Fig. 2).
Fig. 5(a), the color for the right vertical axis for CO saturation ratio is oversaturated, I suggest changing to another color.
The legend of Figure 8 is not very clear.

Reply: We have made all suggested modifications in the figures in the revised manuscript.

[Figure]

Figure 2: Mean CO surface concentrations (± standard error estimate) at Stations 1–14 measured on 25 May 2021 (blue bars) and on 26 May 2021 (orange bars).

[Figure]

Figure 5: (a) Concentrations of dissolved CO (CO$_{surf}$; filled orange dots) and CO saturation ratios (α; filled light blue squares) at all stations in the Ria Formosa Lagoon on 25 May (Day1) and 26 May 2021 (Day2).

[Figure]

Figure 8: The CDOM absorption spectra for different groups (Initial 0, Light-24h-A, Light-24h-B, Light-48h-A, Light-48h-B) of the aquaculture photo-incubations (The curves corresponding to different groups are their fitted curves with the same color).

---

## Author Comment (AC3)

We are thankful to Referee 2 for the thoughtful and constructive comments. According to these suggestions, we have supplemented several references and made corresponding corrections/additions to the manuscript. Hereby are our detailed responses:

The authors report a data-set of dissolved CO concentration at 14 stations in Ria Formosa Lagoon collected on 25 and 26 May 2021 (n=28). The data are discussed in relation to basic environmental variables (temperature, salinity, FDOM) measured with a EXO-2 YSI probe. They made one incubation of CO photo-production.

The CO data were corrected by a factor of 3 due to analytical problem with the CO Gas Analyzer (that was not specified). This correction factor was computed by comparing atmospheric CO with data from atmospheric monitoring stations Mace Head (Ireland) and Terceira (Azores). While it is laudable that the authors are transparent about this correction, I think this is a problem because CO concentrations are quite close to equilibrium (saturation ratio on average of <8) so uncertainty on the dissolved concentration propagates to a large error on the saturation ratio.

Reply: Thank you for pointing out this issue. We acknowledged that the issue stems from a malfunction in the calibration system. There seemed to be errors with the calibration algorithm: The CO analyzer did not report any error during the warm-up progress including self-test after power-on, indicating that the detector, carrier gas pressure, and lamp were working normally. We repeated calibrations over time, using the same standard gases, to ensure that the detector consistently responds to the point calibration. The results were consistent and reproducible. However, the values obtained from the analyzer during the calibration were only about one-third of those given for the standard gases. Further, we compared the average atmospheric CO mole fraction for the same month (May 2021) from atmospheric monitoring stations in Mace Head and Teixeira. A correction factor f (3.12) was determined. This was the fundamental reason for using the correction factor f for the CO analyzer measurements.

Besides this, both water and atmospheric samples were all corrected by this factor, so the overall relative error in the saturation ratios was not affected. Moreover, our conclusion that the lagoon acted as a source of CO to the atmosphere is not affected by the correction.

The authors derived the FDOM data from the YSI EXO-2 when they had access to a UV-Vis Spectrophotometer (as stated) so they could have made spectrophotometric measurements of CDOM, with little extra consumables and workload. This would have been much more precise because my experience with the YSI EXO-2 is that the FDOM sensor tends to provide erratic measurements due to the interaction of scattering light on the sensors and presence of suspended particles in water. I would expect in a lagoon environment quite a lot of these interferences, I would not trust the FDOM data from the sensor.

While the Chl-*a* sensor of the YSI EXO-2 performs better than FDOM sensor, it is still not optimal (compares poorly with discrete Chla measurements), and again the authors could have gone into the trouble of measuring a relatively basic variable such as Chla concentration on 28 samples. The Chla is not just a descriptive variable but is used to calculate the CO production by phytoplankton in the mass budget.

The production of CO in the dark was only estimated on one station at the effluent of the aquaculture facility. While I see from the M&M that these incubations require some work, I think it could have been feasible to make more of these incubations in different water masses of the Ria Formosa Lagoon, since the authors had all of this apparatus and equipment on site.

In conclusion, I find that the sampling effort of CO concentrations was "light" (only 14 stations during 2 consecutive days). It could have been useful to make surveys during low tide, or during other seasons. If the production of CO is mostly due to photo-production, it would have made sense to make night-day cycles. Indeed, CO concentrations change by a factor of 5 during night-day oscillations (Ohta 1997), so potentially a much stronger signal that the variability across the data-set.

Reply: Thank you for your insightful comments and observations regarding our data collection methods. We value and appreciate your expertise on the matter.

Regarding using the YSI EXO-2 sensor and its limitations in lagoon environments, we acknowledge the challenges associated with light scattering and suspended particles, which can potentially interfere with the readings. We are also aware of the benefits of using a UV-Vis Spectrophotometer for FDOM and chlorophyll-*a* measurements. However, this study was conducted under the ASSEMBLE+ Transnational Access Program, which is a short-term collaboration. Given the constraints and limited time available, we prioritized using available equipment to maximize the data collection within the given time frame.

Although the YSI-EXO sensor might not be the most optimal choice for analytical measurements, its application in our study was deemed fit for purpose. We would like to highlight that the YSI-EXO sensor is routinely used by CCMAR in its lagoon (Jacob et al., 2020; Cravo et al., 2022; Caetano et al., 2023). These studies have found the YSI-EXO sensor to provide valuable insights into lagoon dynamics and have successfully utilized the data in their analyses.

Regarding the Chl-*a* measurements, we understand the importance of precision, especially given its use in calculating CO production by phytoplankton. We acknowledge the potential variability between sensor and discrete measurements. However, for the scope and purpose of this specific investigation, the YSI-EXO Chl-*a* sensor provided us with the necessary data to draw our conclusions (e.g., Chl-*a* concentrations decreased from the Ramalhete to the Faro–Olhão inlet (Aníbal et al., 2019), in correspondence to increasing distance from the nutrient-rich plumes from the WWTP). Certainly, if the diurnal variation could be taken into account in the subsequent fieldwork investigation, the RF lagoon CO cycle would be more comprehensively deciphered.

Minor comments

In L200: Please note that the saturation ratio is proportional to the CO concentration because the saturation ratio is computed from the CO concentration, so this finding is meaningless; refer to Berges (1997).

Reply: Thank you for pointing out this issue. To clarify our statement, we modified it. It now reads, 'The waters of the Ria Formosa Lagoon were consistently supersaturated with CO at all stations, with saturation ratios ranging from 1.7 to 32.2 (mean ± SD: 7.7 ± 5.9; Fig. 5a).'

The authors mention they took samples in the late afternoon (~17:00h local time), so well after the maximum of daily irradiance (when CO would have been expected to be max. I imagine that the 14 stations spread over a distance of about 4km were not sampled instantaneously. Can you specify the time of sampling?

Reply: The sampling time on the 25th and 26th of May was around 17:00~18:38 and 16:00~17:35, respectively, and samples were processed immediately after that. Considering the limited time frame in which the sampling was conducted, we would not expect significant differences in CO. For clarity, this information is added to the revised version of the manuscript.

The sample abbreviation (T) is used for temperature either in °C or K

Reply: We have replaced the 'water temperature (T)' with 'water temperature (T in Celsius degrees)'.

The authors should explain why they sampled in quartz glass bottles?

Reply: We used quartz glass bottles two times: 1) for CO concentration measurements (see Section 3.1) and 2) for the incubation experiment (see Section 3.4). We used quartz glass bottles for the incubation experiment to ensure the highest light transmittance possible (also in the UV-B range). To inform the reader about this, we have added a more detailed description of the vials used: (DURAN®, quartz glass, GL 45, DWK Life Sciences, Germany).

In fact, for the measurements, we should have used opaque vials for the concentration measurements. In case to avoid the extra CO photoproduction in the vials, we stored the vials immediately in the dark after sampling. During the headspace pretreatment, the vials were covered with opaque plastic.

The authors should explain how the quartz glass bottles were closed gas-tight. With stoppers?

Reply: Bubble-free water samples were sealed with rubber stoppers and aluminum caps to determine dissolved CO. We added an explanation to the revised manuscript.

**References**

Aníbal, J., Gomes, A., Mendes, I., and Moura, D.: Ria Formosa: challenges of a coastal lagoon in a changing environment, http://hdl.handle.net/10400.1/12475, 2019.

Caetano, S., Correia, C., Vidal, A. F. T., Matos, A., Ferreira, C., and Cravo, A.: Fate of microbial contamination in a South European Coastal Lagoon (Ria Formosa) under the influence of treated effluents dispersal. Journal of Applied Microbiology, 134(8), lxad166, https://doi.org/10.1093/jambio/lxad166, 2023.

Cravo, A., Barbosa, A. B., Correia, C., Matos, A., Caetano, S., Lima, M. J., and Jacob, J.: Unravelling the effects of treated wastewater discharges on the water quality in a coastal lagoon system (Ria Formosa, South Portugal): Relevance of hydrodynamic conditions. Marine Pollution Bulletin, 174, 113296, https://doi.org/10.1016/j.marpolbul.2021.113296, 2022.

Jacob, J., Correia, C., Torres, A. F., Xufre, G., Matos, A., Ferreira, C., ... and Cravo, A.: Impacts of decommissioning and upgrading urban wastewater treatment plants on the water quality in a shellfish farming coastal lagoon (Ria Formosa, South Portugal). Journal of Coastal Research, 95(SI), 45-50, https://doi.org/10.2112/SI95-009.1, 2020.